# IDP: Iterative Differentiable Pruning for Neural Networks with Parameter-free Attention

## Abstract

Deep Neural Network (DNN) pruning is an effective method to reduce the size of a model, improve the inference latency, and minimize the power consumption on DNN accelerators, at the risk of decreasing model accuracy. In this paper, we propose a novel differentiable pruning scheme, Iterative Differentiable Pruning or IDP which offers state-of-the-art qualities in model size, accuracy, and training cost. IDP creates soft pruning masks based on fixed-point attention for a given sparsity target to achieve the state-of-the-art trade-offs between model accuracy and inference compute with negligible training overhead. We evaluated IDP on various computer vision and natural language processing tasks, and found that IDP delivers the state-of-the-art results. For MobileNet-v1, IDP can achieve 68.2% top-1 ImageNet1k accuracy with 86.6% sparsity which is 2.3% higher accuracy than the latest state-of-the-art pruning algorithms. For ResNet18, IDP offers 69.5% top-1 ImageNet1k accuracy with 85.5% sparsity at the same training cost which is 0.8% better than the state-of-the-art method. Also, IDP demonstrates over 83.1% accuracy on Multi-Genre Natural Language Inference with 90% sparsity for BERT, while the next best from the existing techniques shows 81.5% accuracy.

## 1 Introduction

While advanced deep neural networks (DNN) have exceeded human performance on many complex cognitive tasks (Silver et al., 2018), their deployment onto mobile/edge devices, such as watches or glasses, for enhanced user experience (i.e., reduced latency and improved privacy) is still challenging. Most such on-device systems are battery-powered and are heavily resource-constrained, thus requiring DNNs to have very high power/compute/storage efficiency (Wang et al., 2019; Wu et al., 2018; Howard et al., 2017; Vasu et al., 2022; Wang et al., 2020b).

Such efficiency can be accomplished by mixing and matching various techniques, such as designing efficient DNN architectures like MobileNet/MobileViT/ MobileOne (Sandler et al., 2018; Mehta & Rastegari, 2022; Vasu et al., 2022), distilling a complex DNN into a smaller architecture (Polino et al., 2018), quantizing/compressing the weights of DNNs (Cho et al., 2022; Han et al., 2016; J. Lee, 2021; Park & Yoo, 2020; Li et al., 2019; Zhao et al., 2019), and pruning near-zero weights (Peste et al., 2021; Kusupati et al., 2020; Liu et al., 2021; Zhang et al., 2022; Sanh et al., 2020; Zafrir et al., 2021; Zhu & Gupta, 2018; Wortsman et al., 2019). Also, pruning is known to be highly complementary to quantization/compression (Wang et al., 2020b) when optimizing a DNN model. Training a larger model and then compressing it by pruning has been shown to be more effective in terms of model accuracy than training a smaller model (Li et al., 2020) from the beginning. However, pruning comes at the cost of degraded model accuracy, and the trade-off is not straightforward (Kusupati et al., 2020).

Hence, a desirable pruning algorithm should achieve high accuracy and accelerate inference for various types of networks without significant training overheads in costs and complexity. In this work, we propose a simple yet effective pruning technique, Iterative Differentiable Pruning or IDP based on an parameter-free attention mechanism (Bahdana et al., 2015; Xu et al., 2015) that satisfies all of the above criteria. Our attention approach allows a pruning mask to be differentiable, and lets training-loss decide whether/how each weight will be pruned. Therefore, such a loss-driven differentiable pruning mask will help capture the interactions among weights automatically without expensive mechanisms (Liu et al., 2021). Also, IDP requires neither additional learning parameters (Zhang et al., 2022) nor complicated training flows (Peste et al., 2021), yet offers a precise control on the target

| | STR[a] | GraNet[b] | OptG[c] | ACDC[d] | MVP[e] | POFA[f] | IDP |
|---|---|---|---|---|---|---|---|
| Accuracy on Conv-Nets | ✓ | ✓ | ✓✓ | ✓✓ | ? | ? | ✓✓✓ |
| Accuracy on Transformers | ✓ | ? | ✓ | ? | ✓✓ | ✓✓ | ✓✓✓ |
| Inference speed (MAC) | ✓✓✓ | ✓ | ✓✓✓ | ✓ | ? | ? | ✓✓ |
| Training speed | ✓✓✓ | ✓✓ | ✓✓ | ✓ | ? | ? | ✓✓ |
| Training stability | ✓ | ✓✓ | ✓✓✓ | ✓✓✓ | ? | ? | ✓✓✓ |
| Training flow | simple | complex | complex | complex | complex | complex | simple |
| Extra training parameters | a few | none | many | none | many | none | none |

[a] Soft threshold weight re-parameterization for learnable sparsity (Kusupati et al., 2020).
[b] Sparse training via boosting pruning plasticity with neuro–regeneration (Liu et al., 2021).
[c] Optimizing gradient-driven criteria in network sparsity: Gradient is all you need (Zhang et al., 2022).
[d] AC/DC: alternating compressed/decompressed training of deep neural networks (Peste et al., 2021).
[e] Movement Pruning: Adaptive Sparsity by Fine-Tuning (Sanh et al., 2020).
[f] Prune Once for All: Sparse Pre-Trained Language Models (Zafrir et al., 2021).

**Table 1:** Comparison of the state-of-the-art pruning schemes and IDP: IDP can explore a good trade-off bewteen accuracy and inference speed without introducing new learnable parameters and with a simple/fast training flow.

sparsity level (Kusupati et al., 2020) and pushes the state-of-the-art in pruning. Table 1 compares IDP with the latest state-of-the-art pruning schemes, and our major contributions include:

- A differentiable and parameter-free pruning algorithm based on attention.

- Efficiently pruning to offer a high-quality model for a given pruning target.

- The state-of-the-art results on both computer vision and natural language tasks.

## 2 RELATED WORKS

**Trade-offs in Pruning**: Pruning in DNN incurs a complex trade-off between model accuracy and inference speed in terms of MAC (mult-add operations) (Kusupati et al., 2020). A weight can contribute differently to the model accuracy, depending on the number of times it used for prediction (i.e., a weight in convolution filter for a large input) and the criticality of the layer it belongs to (i.e., a weight in a bottleneck layer). Please see Fig. 7 and Section B in Appendix for details. Therefore, even if two models are pruned to the same level, the accuracy and inference speed of each can be vastly different, which makes exploring the best trade-off challenging yet crucial in DNN pruning.

**Unstructured Pruning**: Unstructured schemes make individual and independent pruning decision for each weight to maximize the flexibility and minimize the accuracy degradation. Simple and gradual/iterative pruning based on the weight magnitude has been studied extensively (Zhu & Gupta, 2018; Gale et al., 2019; Frankle & Carbin, 2019; Han et al., 2015). In these schemes, once a weight is pruned, it does not have the *second* chance to become unpruned and improve the model quality. To address such challenges, RigL (Evci et al., 2020) proposes to grow a sparse network by reallocating the removed weights based on their dense gradients. Applying brain-inspired neurogeneration (i.e., unpruning some weights based on gradients) and leveraging pruning plasticity is proposed (Liu et al., 2021). Altering the phase of dense and sparse training to accomplish co-training of sparse and dense models is studied, which results in good model accuracies on vision tasks (Peste et al., 2021). Unlike other magnitude-driven pruning, supermask training (Zhou et al., 2019) integrated with gradient-driven sparsity is proposed in (Zhang et al., 2022), where accumulated gradients are used to generate binary masks and straight-through estimator (Bengio et al., 2013) is used for backward propagation. Motivated by the lottery hypothesis (Frankle & Carbin, 2019), pruning in one-shot based on heuristics (Tanaka et al., 2020) or gradient-driven metrics (Wang et al., 2020a) is explored.

**Structured Pruning**: Unstructured pruning limits inference latency speedup as it takes significant overhead to fetch irregular non-zero index matrices, suffers from poor memory access performance, and does not fit well on parallel computation (Anwar et al., 2017; Liu et al., 2022). Therefore, recent research extends unstructured pruning by imposing a particular sparsity pattern during pruning at the cost of lower model predictive power, but increases the hardware performance during inference. One popular and effective form of structured pruning is channel pruning, where some channels with negligible effects on the model accuracy are discarded (He et al., 2017; Li et al., 2017; Kang & Han, 2020). Using regularization to prune weights in a block is proposed in (Lagunas et al., 2021). N:M pruning enforces that there are N zero weights out of every consecutive M weights (Zhou et al., 2021).

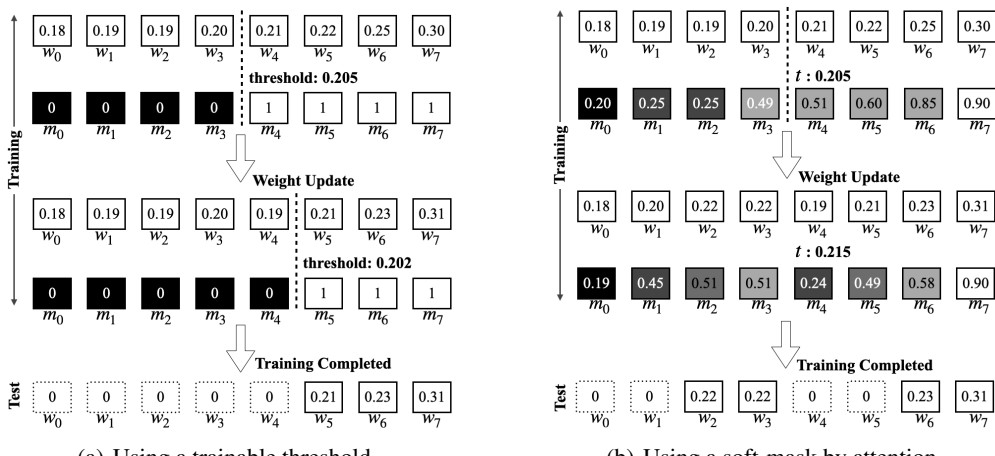

(a) Using a trainable threshold.  (b) Using a soft-mask by attention.

**Figure 1:** Unlike learning a threshold parameter for differentiable pruning (Kusupati et al., 2020; Ning et al., 2020) in (a), IDP generates a soft-mask using parameter-free attention to accomplish differentiable pruning as in (b). The moment a weight is pruned in (a), the weight will not get any update as the mask zeros out the gradient, depriving the opportunity for better model training. However, the schemes in (a) ensure the training and test behaviors are identical, yet makes the sparsity control difficult. While IDP reveals different behaviors during training/test times because the test needs hard-masks, our attention-based soft-mask allows a weight to recover from being pruned over the time, which yields higher model accuracies according to our results.

Such hard constraints make it possible to have a compact memory layout and efficient inferences on hardware (Jeff Pool, 2021; Mishra et al., 2021; Zhou et al., 2021).

**Differentiable Pruning**: Using differentiable techniques for pruning gained popularity due to the advances in network architecture search methods, as well as network compression techniques (Liu et al., 2019; Chang et al., 2019; Cho et al., 2022). Using the first derivative of foothill function (Belbahri et al., 2019) as a mask function and applying L1 regularization on the pruning thresholds has been proposed for differentiable pruning (Ramakrishnan et al., 2020). However, it is not clear why the foothill function is the right choice (i.e., the mask value can exceed 1.0 and is not monotonic). DSA (Ning et al., 2020) finds the layer-wise pruning ratios in a differentiable fashion by computing a channel-wise keep probability drawn from the Bernoulli distribution. DSA requires solving a subproblem in order to keep the expected sparsity ratio satisfied. Applying a differentiable technique for channel pruning was proposed (Kang & Han, 2020) based on the operation-specific observation that a feature map with a large and negative batch mean will get deactivated (or zeroed) by ReLU activation function. A differentiable Markov process is studied for channel pruning, where a state represents a channel and the transitions from states accounts for the probability of keeping one channel unpruned under the condition that the other channel is also retained (Guo et al., 2020). AutoPrune (Xiao et al., 2019) introduces trainable auxiliary parameters (controlled by a regularizer) to generate a hard pruning mask differentiably using STE. Optimizing and training a per-layer pruning threshold with ReLU based on a dynamic re-parameterization method was proposed to allocate the sparsity across all layers (Kusupati et al., 2020), which shows competitive model accuracy and low inference overhead. IDP is a novel kind of differentiable pruning powered by attention-based soft-mask to optimize individual weight masks per task-loss.

## 3    IDP: ITERATIVE DIFFERENTIABLE PRUNING

In this section, we introduce IDP by first providing the key ideas and insights behind IDP in Section 3.1, explain the details in Section 3.2, followed by the training flow with IDP in Section 3.3.

### 3.1    OVERVIEW

Existing differentiable schemes focus on differentiable pruning budget allocation by training a differentiable pruning threshold based on marginal loss or L1-regularization (Kang & Han, 2020; Ning et al., 2020). Meanwhile IDP directly generates an a soft pruning mask based on parameter-free

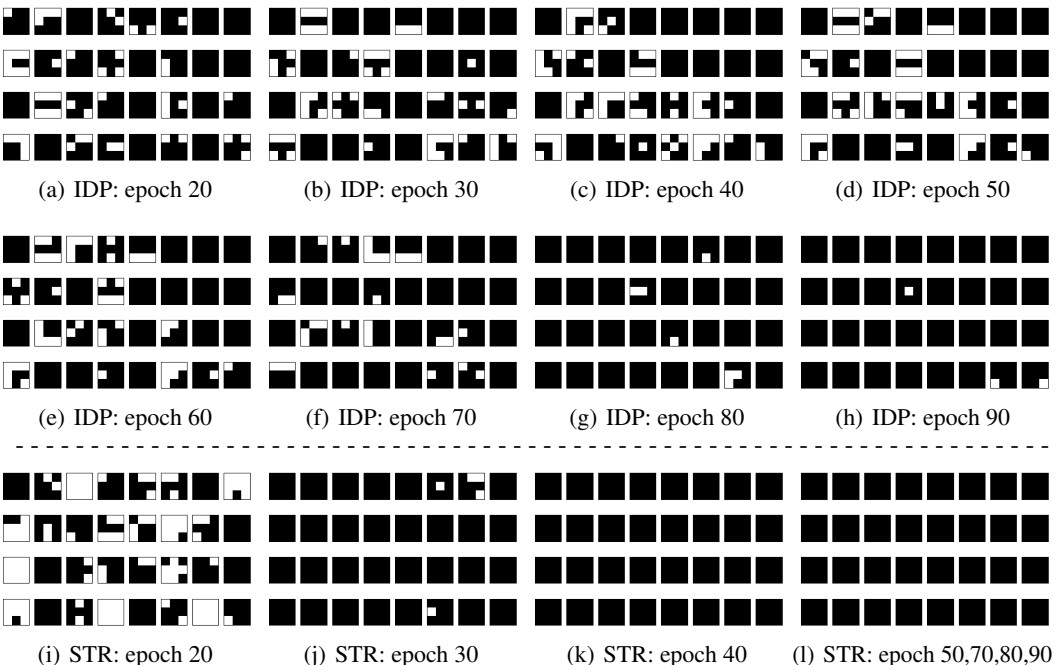

(a) IDP: epoch 20    (b) IDP: epoch 30    (c) IDP: epoch 40    (d) IDP: epoch 50

(e) IDP: epoch 60    (f) IDP: epoch 70    (g) IDP: epoch 80    (h) IDP: epoch 90

(i) STR: epoch 20    (j) STR: epoch 30    (k) STR: epoch 40    (l) STR: epoch 50,70,80,90

**Figure 2:** The effects of IDP and STR (Kusupati et al., 2020) (from Table 1) for the first 3x3 Conv2d layer with 32 filters in MobileNet-v1 on ImageNet1k where each small rectangle represents one 3x3 kernel: For IDP, during the entire end-2-end training, we temporarily round the soft-mask value to make the pruning decisions. The white cell indicates such pruning decision for the corresponding weight has been flipped at least once during the particular epoch, and the black cell indicates the other case.

attention for each weight to make the pruning process differentiable. Fig. 1 illustrates the differences with an example. Learning a pruning threshold is useful for global pruning budget allocation, as the threshold gets adjusted per the task loss as in Fig. 1 (a), but has the following drawbacks:

- Difficult to control a sparsity level, as the pruning threshold is not explicitly related to the sparsity because masks and the thresholds are not directly co-optimized. In Fig. 1 (a), after one weight update, the threshold is reduced from 0.205 to 0.202, but the sparsity is increased from 50% to 66% as $w_4$ becomes smaller than the threshold.

- Once a weight gets pruned during training, it does not get updated as the gradient is masked out. The bottom 4 weights $w_{\{0,1,2,3\}}$ get no weight update due to the zero masks, and can remain likely pruned through the end of training.

In the contrary, IDP allows all the weights to get updated through attention-based soft-mask as in Fig. 1 (b), providing higher flexibility and recovery from undesirable pruning (Guo et al., 2016). For example, consider the weight $w_2$ of a value 0.19 which would have been permanently pruned in (a). IDP discourages a weight from being permanently pruned through the weight update, as still a scaled-down gradient is available. If a near-zero weight continues to get negative gradients over the time (even if it is scaled down by the soft-mask), it can eventually get unpruned at the end of training. In the same way, if a very large weight gets positive gradients many times enough to be near-zero, it will get pruned eventually, even if it was not pruned in the early stage. Hence, such gradual pruning decision over the time during training allows IDP to make better pruning decisions w.r.t. the task loss.

Such *soft* masking allows pruning decisions to be flipped multiple times during the entire training process, and such effects are compared between IDP and STR (Kusupati et al., 2020) (from Table 1) in Fig. 2 where the pruning decisions for the first Conv2d (3x3 kernels with 32 output channels) in MobileNet-v1 for the end-to-end ImageNet1k training are captured. The white cell means that the pruning decision has been flipped (from to-prune to not-to-prune or vice-versa), while the black cell means the decision has never changed, during the given epoch. Then, we can observe from Fig. 2

---

**Algorithm 1** Computing a pruned weight matrix with IDP

---

1: **procedure** PRUNEMASK($W$,$r$)
2:     $W_h = topK(abs(W), (1 - r) \cdot n(W))$
3:     $W_l = topK(-abs(W), r \cdot n(W))$
4:     $t = detach\_autograd(0.5\{min(W_h) + max(W_l)\})$     *//detached from autograd*
5:     $[Z, M] = softmax(\frac{[t^2 \mathbb{J}, W \circ W]}{\tau})$     *//element-wise operation*
6:     **return** $M$
7: **end procedure**

---

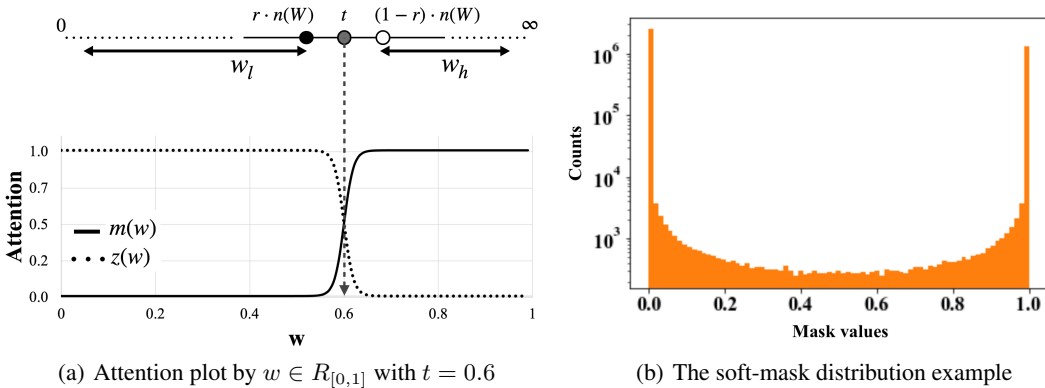

(a) Attention plot by $w \in R_{[0,1]}$ with $t = 0.6$      (b) The soft-mask distribution example

**Figure 3:** Computing $z(w), m(w)$ for the attentions to $Z$ and $M$ with $t$ for the equal attention to $Z$ and $M$.

that while IDP in (a)-(h) continues to flip the decisions until the very late training stage, STR in (i)-(l) mostly finalizes the pruning decisions early (i.e., pruning decision gets static after about 30 epochs).

## 3.2   SOFT PRUNIG MASK WITH PARAMETER-FREE ATTENTION

Instead of a hard-mask, IDP uses a soft-mask generated by attention to the two symbolic states, **"to-prune"** noted as $Z$ or **"not-to-prune"** noted as $M$ for a given weight $w$. Unlike the existing attention-based model optimizations such as (Cho et al., 2022), where the attention anchors are numerically defined, the attention to symbolic states is not straightforward to compute (i.e., Euclidean distance to $Z$ is not available directly), and thus requires a different approach.

Let us imagine continuous and differentiable functions, $z, m : \mathbb{R}_{[0,\infty]} \mapsto \mathbb{R}_{[0,1]}$ where $z$ and $m$ are the attentions to $Z$ and $M$, respectively. Then, $z$ and $m$ must satisfy the following conditions to be used as attention-based soft-mask for the magnitude-driven weight pruning:

- $z(|a|) < z(|b|)$ for any $|a| > |b|$: a weight with smaller magnitude has more attention to $Z$.
- $m(|a|) > m(|b|)$ for any $|a| > |b|$: a weight with larger magnitude has more attention to $M$.
- $z(w) + m(w) = 1$ for any $w$: the total attention is 1.

Accordingly, due to the monotonicity and continuity, there must exist $t \in \mathbb{R}_{\geq 0}$ such that $z(t) = m(t) = 0.5$ for the equal attention to $Z$ and $M$, which leads to the following boundary conditions:

$$z(w) = \begin{cases} 1 & \text{if } w = 0 \\ 0 & \text{if } |w| = \infty \\ \frac{1}{2} & \text{if } |w| = t \end{cases}$$

One can use any function that satisfies all of the above conditions, and take $m(w)$ as a soft-mask of $w$ for train-time pruning. In IDP, we uniquely identify $t$ for a given prune ratio $r \in [0, 1)$ for a layer with a weight matrix $W$ as follows based on the the pruning context:

$$W_h = topK(abs(W), (1 - r) \cdot n(W)) \tag{1}$$

$$W_l = topK(-abs(W), r \cdot n(W)) \tag{2}$$

---

**Algorithm 2** Training flow for IDP

---

1: **procedure** TRAIN($\epsilon, s, r, W = [W_0, W_1, ...]$)
2:     **for** epoch $e$ in $[0, 1, 2, s)$ **do**
3:         **for** each mini-batch **do**
4:             forward with $[W_0, W_2, ...]$, backward and weight update
5:         **end for**
6:     **end for**
7:     $W_p = topK(-abs(W), r \cdot n(W))$
8:     $[r_0, r_1, ...] = [\frac{n(W_p \cap W_0)}{n(W_0)}, \frac{n(W_p \cap W_1)}{n(W_1)}, ...]$
9:     **for** epoch $e$ in $[s, s+1, s+2, ...]$ **do**
10:         $[\hat{r_0}, \hat{r_1}, ...] = min(1, \epsilon \cdot (e-s)) \cdot [r_0, r_1, ...]$
11:         **for** each mini-batch **do**
12:             **for** $i \in \{0, 1, ...\}$ **do**z
13:                 $M_i$= PruneMask($W_i, \hat{r_i}$)
14:                 $\hat{W_i} = M_i \circ W_i$    *//element-wise operation*
15:             **end for**
16:             forward with $[\hat{W}_0, \hat{W}_1, ...]$, backward and weight update
17:         **end for**
18:     **end for**
19:     $W_i = \lfloor M_i \rceil \circ W_i, \forall i \in \{0, 1, ..\}$
20: **end procedure**

---

where $topK(X, k)$ is selecting the largest $k$ elements from a matrix $X$, $abs(X)$ is an element-wise absolute operation, and $n(X)$ returns the number of elements. Then, we compute the following:

$$t = 0.5\{min(W_h) + max(W_l)\} \tag{3}$$

Note that Eq. (1,2,3) compute outside the autograd (i.e., do not participate the backward-propagation using *torch.no_grad()*), as only $t$ as a scalar is required for the next step. Then, we propose the following differentiable parameter-free attention to compute $z(m)$, $m(w)$, and the soft-pruned weight $\hat{w}$, w.r.t. $t$ which represents the current weight distribution.

$$\hat{w} = m(w) \cdot w = [z(w), m(w)] \cdot [0, w]$$
$$= softmax(\frac{concat(t^2, w^2)}{\tau}) \cdot [0, w] = softmax(\frac{[t^2, w^2]}{\tau}) \cdot [0, w] \tag{4}$$

where $\tau$ is the temperature parameter (see Section C in Appendix for more discussions) and $m(w)$ is directly used as a pruning mask during training. Fig. 3 (a) illustrates how $t$ is obtained in IDP and a soft-mask is computed: essentially, $t$ in gray, is in the exact middle of the largest pruned weight in black and the smallest unpruned weight in white when a hard-mask is applied for a given sparsity ratio $r$, thus giving us the equal attention to $Z$ and $M$. Fig. 3 (b) shows the soft-mask distribution of the word-embedding layer of BERT (Devlin et al., 2019). Therefore, Eq. (4) satisfies all the constraints for $z$ and $m$. Further, the softmax is fast enough to process a large number of weights. The soft-mask matrix generation by IDP is stated in Algorithm 1, which has $O(W)$ runtime complexity.

In comparison to scaled-dot product attention (or QKV attention) in Transformer (Vaswani et al., 2017), our parameter-free attention has the following similarity, differences, and advantages:

- Our parameter-free attention in Eq. (4) has structure-wise similarity with QKV attention: $[0, w]$ corresponds to $V$, and $t$ and $w$ maps to Q and K, respectively, except that we use a simple *concat* instead of dot product to get the attention scores from *softmax*.

- Our attention does not have learnable attention parameters, based on the structure of pruning. Instead, we let weights learn and improve their distribution to generate better attention scores (through weight update) in the presence of our soft-mask for the subsequent iterations.

- Having no extra parameters and using *concat* makes IDP very efficient in terms of training speed and keeps training flow simple, yet our experimental results show its effectiveness.

In short, using a soft-mask by $m$ enables the pruning decision for the weights around the pruning boundary (i.e., $t$) to be driven by a task loss through back-propagation rather than the weight value itself. On the contrary, a pruned weight will get zero gradient with a hard-mask once pruned.

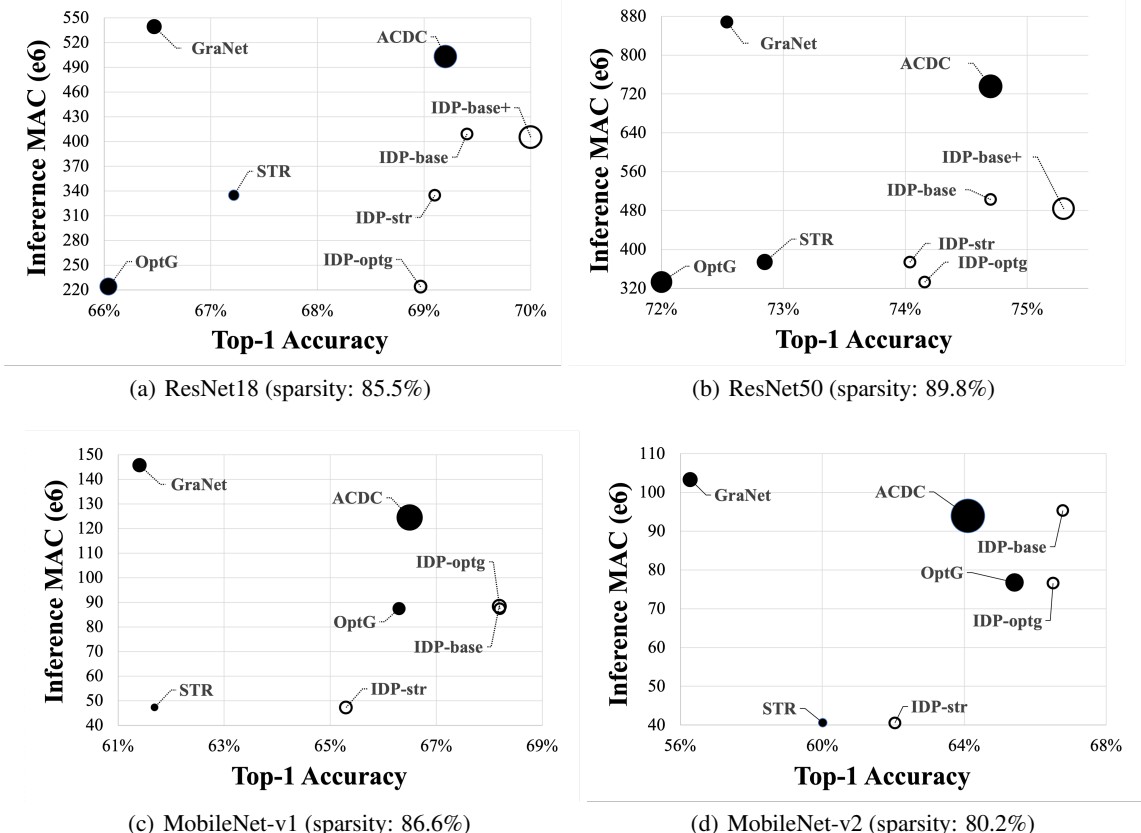

(a) ResNet18 (sparsity: 85.5%)

(b) ResNet50 (sparsity: 89.8%)

(c) MobileNet-v1 (sparsity: 86.6%)

(d) MobileNet-v2 (sparsity: 80.2%)

**Figure 4:** IDP-powered pruning (in white box markers) delivers the Pareto superiority to the other schemes (i.e., the top-bottom corner is the best trade-off) for ResNet18, ResNet50, and MobileNet-v1/v2 on ImageNet1k. The size of markers indicates the relative training overheads. The detailed numbers are in Table. 2.

### 3.3 IDP ALGORITHM AND TRAINING FLOW

In order to obtain $t$ in Eq. (3), IDP needs a target pruning ratio $r$. The pruning ratio can be computed by selecting the top weights with larger magnitudes across all the layers and then instantly convert the selections into the per-layer ratios. Another way is to handcraft per-layer ratios, or reuse an existing configuration. Also, IDP is using the softmax operation which makes the *softness* concentrated over the weights around the $t$ (as shown in Fig. 3). Hence, we gradually increase the target pruning ratio from 0 to $r$ so that all low magnitude weights in the pruning range have a chance to use a soft-mask and settle down smoothly. For that purpose, we introduce a scaling step $\epsilon$ to let each weight have opportunities to leverage a soft-mask at least once, which leads to the training flow in Algorithm 2.

In lines 2-6, a normal training is performed for the first $s$ epochs. Then, in lines 7 and 8, the per-layer target pruning ratio is computed by selecting the bottom $r \cdot n(W)$ weights globally in terms of the magnitude. Then, in the remaining epochs, we use IDP to generate soft-masks as in the line 13, while gradually increasing the target ratio as in lines 9 and 10. Once the entire training is over, we binarize the last mask for each weight to output the fully pruned weight for inference as in the line 19.

### 4 EXPERIMENTAL RESULTS

We compared our **IDP** with state-of-the-art unstructured pruning schemes on various computer vision and natural language models. We used two x86 Linux nodes with eight NVIDIA V100 GPUs each.

**Vision Benchmark:** We compared the proposed **IDP** with the latest prior arts, **STR** (Kusupati et al., 2020), **GMP** (Zhu & Gupta, 2018), **DNW** (Wortsman et al., 2019), **GraNet** (Liu et al., 2021),

| Network Sparsity | Method | Top-1 (%) | GPU$^\$$ hours | MAC ($\times e6$) | Network Sparsity | Method | Top-1 (%) | GPU$^\$$ hours | MAC ($\times e6$) |
|---|---|---|---|---|---|---|---|---|---|
| | Dense | 69.8 | 91 | 1814.1 | | Dense | 76.1 | 135 | 4089.2 |
| | GMP | 65.2 | 118 | 263.5 | | GMP | 73.6 | 263 | 419.0 |
| | DNW | 64.4 | 112 | 263.5 | | DNW | 70.7 | 254 | 419.0 |
| | GraNet* | 66.0 | 108 | 539.6 | | GraNet* | 72.5 | 175 | 868.0 |
| ResNet18 | STR | 66.7 | 93 | 334.6 | ResNet50 | STR | 72.8 | 227 | 373.7 |
| 85.5% | OptG* | 65.5 | 151 | 223.7 | 89.8% | OptG* | 72.1 | 322 | 333.0 |
| | ACDC | 68.7 | 194 | 502.8 | | ACDC | 74.7 | 346 | 735.6 |
| | IDP-base | 69.0 | 92 | 408.6 | | IDP-base | 74.7 | 177 | 502.8 |
| | IDP-base+ | **69.5** | 183 | 405.1 | | IDP-base+ | **75.3** | 332 | 483.0 |
| | IDP-str | 68.6 | 92 | 334.7 | | IDP-str | 74.0 | 177 | 373.7 |
| | IDP-optg | 68.5 | 95 | 223.7 | | IDP-optg | 74.2 | 181 | 332.9 |
| | Dense | 70.9 | 151 | 568.7 | | Dense | 71.9 | 162 | 300.8 |
| | GraNet* | 61.4 | 200 | 145.7 | | GraNet* | 56.3 | 239 | 103.4 |
| MobileNet-v1 | STR | 61.7 | 96 | 47.2 | MobileNetv-2 | STR | 60.0 | 155 | 40.6 |
| 86.6% | OptG* | 66.3 | 185 | 87.4 | 80.2% | OptG* | 65.4 | 297 | 76.8 |
| | ACDC | 66.5 | 349 | 124.5 | | ACDC | 64.1 | 442 | 93.9 |
| | IDP-base | **68.2** | 153 | 88.3 | | IDP-base | **66.8** | 193 | 95.3 |
| | IDP-str | 65.3 | 167 | 47.2 | | IDP-str | 60.7 | 167 | 40.6 |
| | IDP-optg | 68.2 | 162 | 87.3 | | IDP-optg | 66.5 | 187 | 76.6 |

$^\$$ the total GPUs hours on all the GPUs in synchronous distributed training.
$^*$ used only one with 8 GPUs due to the limitations in the public code.

**Table 2:** IDP compared with other unstructured pruning algorithms on with ImageNet1K shows the best trade-off among accuracy, inference MAC, and training overheads. More results are available in Section D in Appendix.

**OptG** (Zhang et al., 2022), and **ACDC** (Peste et al., 2021) on ResNet18, ResNet50, MobileNet-v1, and MobileNet-v2 (He et al., 2016; Howard et al., 2017; Sandler et al., 2018) with the ImageNet1k dataset (Deng et al., 2009). Since all of these schemes have been experimented only with ResNet50 and/or MobileNet-v1, we reproduced the pruning results in our controlled environment with the identical data augmentations by running the official implementations from the authors (Kusupati et al., 2020; Zhang et al., 2022; Peste et al., 2021; Liu et al., 2021) or verified implementations from the prior arts (Wortsman et al., 2019; Zhu & Gupta, 2018) as in Section F in Appendix. Since the primary goal of pruning is to trade-off the model accuracy with the compute reduction as in Section 2, we measured the accuracies and inference-time Multiply-Accumulate Operation (MAC) on each experiment with layer fusion (i.e., BatchNorm folding), and mainly focused on the high-sparsification cases. Note that the MAC is purely theoretical and reported to understand the trade-off among accuracy, size, and compute across various algorithms. In our experiments with ImageNet1k, all layers, including both the first and last layers, are pruned without any restriction. Also, we estimated training efficiency in wall-clock time by measuring the total GPU-hours (Chen et al., 2022), which is crucial to capture the algorithmic complexity and training flow overhead.

We applied not only the proposed hyper-parameters from the authors but also a set of further fine-tuned hyper-parameters for competing methods. Also, since each algorithm used a different number of epochs and showed results at different sparsity levels, **a)** we ran **STR** first to set the target sparsity levels for all the networks for fair comparisons, because all other schemes can control the sparsity level precisely, **b)** we trained ResNet18/50 for 100 epochs and MobileNet-v1/v2 for 180 epochs following (Peste et al., 2021; Kusupati et al., 2020; Liu et al., 2021; Zhang et al., 2022) except **STR** (which diverged with more epochs for MobileNet-v1/v2). For **IDP**, we started pruning from the epoch 16 at the rate of 1.5% of the target sparsity per epoch for all the experiments which correspond to $s = 16$ and $\epsilon = 0.015$ in Algorithm 2. For detailed experiment configurations, please refer to Table 4. Every experiment began with a randomly initialized model (i.e., no pre-trained model). For **IDP**, we had the following variants to demonstrate the value of **IDP** with the same training overhead or with the same per-layer pruning budgets.

- **IDP-base** computes the target sparsity level by using weight magnitude globally, as in the lines 7 and 8 of Algorithm 2.
- **IDP-base+** is the same as **IDP-base** except it runs for additional epochs to consumes the similar total GPU hours with **ACDC**.

| Network | Validation dataset | Sparsity | Methods | | | | | | |
|---|---|---|---|---|---|---|---|---|---|
| | | | Dense | STR* | OptG | GMP | MVP | POFA | IDP |
| Bert-base | matched | 90 | 84.5 | 75.8 | 78.5 | n/a | 81.2 | 81.5 | **83.1** |
| | | 94 | | 74.4 | 76.9 | 74.8 | 80.7 | n/a | **82.0** |
| | mismatched | 90 | 84.9 | 76.3 | 78.3 | n/a | 81.8 | 82.4 | **83.0** |
| | | 94 | | 74.1 | 76.5 | 75.6 | 81.2 | n/a | **82.4** |

\* Since **STR** cannot control the sparsity precisely, we report the metrics with our closest achieved sparsity levels, 85.9% for the 90% case and 91.8% for the 94% case.

**Table 3:** **IDP** delivers the best accuracies on the BERT-based model with the MNLI (Multi-Genre Natural Language Inference) task of the GLUE benchmark.

- **IDP-str/optg** uses the sparsification levels resulted from **STR/OptG** as input to achieve the similar MAC by overwriting $[r0, r1, ...]$ in the line 8 of of Algorithm 2.

Our experimental results are highlighted in Fig. 4, where the size of circles indicates the relative training overhead due to pruning. Note that we used only one single node with 8 GPUs due to the limitation in the official implementations for **GraNet** and **OptG**, thus both have the advantage of not having the inter-node communication cost. Also, each approach imposes a different level of training-time overhead, mainly due to the various complexities of training flow and pruning itself as captured in Fig. 4. Overall results can be summarized as follows:

- **IDP** showed the best the model accuracy: **IDP-base** on ResNet18 delivered 69% Top-1 accuracy which is superior to other schemes but at higher MAC than only **STR** and **OptG**.

- **IDP** offered the better model accuracy for a given pruning target: With the custom sparsification target for each layer, **IDP-str/optg** demonstrated the 2-3% higher Top-1 accuracy at the same MAC, demonstrating the effectiveness of the proposed method.

- When we use the similar GPU hours for additional epochs with **ACDC** which is noted as **IDP-base+**, our method further improved the Top-1 accuracy from 69% to 69.5% for ResNet18 and from 74.7% to 75.3% for ResNet50 with slight fewer MACs.

**NLP Benchmark:** We compared **IDP** with the state-of-the-art pruning results from **MVP** (Sanh et al., 2020) and **POFA** (Zafrir et al., 2021) (quoted from the respective papers) in addition to **OptG** and **STR** (reproduced in our environment) on a BERT model (Devlin et al., 2019) for an NLP task, MNLI (Multi-Genre Natural Language Inference) of the GLUE benchmark. Following the setups in **MVP** and **POFA**, we used the same batch size 32 per GPU (i.e., global mini-batch size of 512), excluded the embedding and the last linear layer from pruning, and trained the *bert-base-uncased* from HuggingFace from scratch with self-distillation. The teacher for the distillation is from the best checkpoint in the first 40 epochs. We use the weights of 0.95 on the distillation loss and 0.05 on the task loss for **IDP**, and 0.75 on the distillation loss and 0.25 on the task loss for **STR**. We could not use distillation for **OptG**, as it caused the out-of-memory error due to the extra-parameter overheads from both pruning and distillation. For more details, please refer to Table 4 in Appendix. Our experimental results in Table 3 can be summarized as follows:

- **STR** underperforms on all the best cases even though it could not achieve the target sparsity.

- **OptG** shows better model accuracy only than **GMP** and worse than others in our setup.

- **IDP** outperformed all other methods for both MNLI validation datasets.

## 5 CONCLUSION

In this work, we proposed a differentiable pruning method, IDP which yields the state-of-the-art pruning quality on popular computer vision and natural language models. Our method requires no additional learning parameters, yet keeps the training flow simple and straightforward, making it a practical method for real-world scenarios. We plan to extend our differentiable pruning into quantization, making both jointly differentiable and optimizable by the task-loss.

## 6    ETHICS STATEMENT

The proposed technique does not incur any concern or penitential issues around ethics.

## 7    REPRODUCIBILITY STATEMENT

We provide the details hyper-parameters and experimental configurations in Seciton 4 and Table 4 in Appendix. For Imagenet1k, we used common data augmentation techniques: RandomResized-Crop(224), RandomHorizontalFlip, and Normalize(mean=[0.485, 0.456, 0.406], std=[0.229, 0.224, 0.225]) based on the prior arts (Park & Yoo, 2020; J. Lee, 2021). During evaluation, we used the following augmentations: Resize(256), CentorCrop(224), and Normalize(mean=[0.485, 0.456, 0.406], std=[0.229, 0.224, 0.225]). For the MNLI benchmark, we used the default setup from HuggingFace.

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

| Network | Method | Batch size | #epochs | #wu | #GPUs | #Nodes | Main optimizer, scheduler | Mask optimizer, scheduler | other params |
|---|---|---|---|---|---|---|---|---|---|
| ResNet18 | GraNet | 128 | 100 | 5 | 8 | 1 | SGD 0.9 1e-4 (0.0 for BN), cosine 0.1 | n/a | init density: 0.5 |
|  | GMP | 128 | 100 | 5 | 16 | 2 | SGD 0.875 1e-5, cosine 0.256 | same as the main | init value: -3200 |
|  | STR | 256 | 100 | 5 | 16 | 2 | SGD 0.875 2.251757813e-5, cosine 0.256 | same as the main |  |
|  | OptG | 256 | 100 | 5 | 16 | 2 | SGD 0.9 1e-4, cosine 0.1 | SGD 0.9 0, cosine 0.1 | $\beta$ : 1.0 |
|  | ACDC | 256 | 100 | 5 | 8 | 1 | SGD 0.875 3.05175781e-4, cosine 0.256 | n/a | 8 alterations |
|  | IDP-base/str/optg | 1024 | 100 | 5 | 16 | 2 | SGD 0.9 1.4e-5, cosine 1.8 | n/a | $\tau$ : $1e^{-4}$ , $\epsilon$ : 0.015 |
|  | IDP-base+ | 1024 | 200 | 5 | 16 | 2 | SGD 0.9 1.4e-5, cosine 1.8 | n/a | $\tau$ : $1e^{-4}$ , $\epsilon$ : 0.015 |
| ResNet50 | GraNet | 128 | 100 | 5 | 8 | 1 | SGD 0.9 1e-4 (0.0 for BN), cosine 0.1 | n/a | init density: 0.5 |
|  | GMP | 128 | 100 | 5 | 16 | 2 | SGD 0.875 1e-5, cosine 0.256 | same as the main | init value: -3200 |
|  | STR | 256 | 100 | 5 | 16 | 2 | SGD 0.875 2.251757813e-5, cosine 0.256 | same as the main |  |
|  | OptG | 256 | 100 | 5 | 16 | 2 | SGD 0.9 1e-4, cosine 0.1 | SGD 0.9 0, cosine 0.1 | $\beta$ : 1.0 |
|  | ACDC | 256 | 100 | 5 | 8 | 1 | SGD 0.875 3.05175781e-4, cosine 0.256 | n/a | 8 alterations |
|  | IDP-base/str/optg | 1024 | 100 | 5 | 16 | 2 | SGD 0.9 1.4e-5, cosine 1.8 | n/a | $\tau$ : $1e^{-4}$ , $\epsilon$ : 0.015 |
|  | IDP-base+ | 1024 | 200 | 5 | 16 | 2 | SGD 0.9 1.4e-5, cosine 1.8 | n/a | $\tau$ : $1e^{-4}$ , $\epsilon$ : 0.015 |
| MobileNet-v1 | GraNet | 128 | 180 | 5 | 8 | 1 | SGD 0.9 1e-4 (0.0 for BN), cosine 0.1 | n/a | init density: 0.5 |
|  | STR | 256 | 100 | 5 | 16 | 2 | SGD 0.875 3.751757813e-5, cosine 0.256 | same as the main | init value: -12800 |
|  | OptG | 256 | 180 | 5 | 16 | 2 | SGD 0.9 4e-4, cosine 0.1 | SGD 0.9 0, cosine 0.1 | $\beta$ : 1.0 |
|  | ACDC | 256 | 180 | 5 | 8 | 1 | SGD 0.875 3.05175781e-4, cosine 0.256 | n/a | 8 alterations |
|  | IDP-base/str/optg | 1024 | 180 | 5 | 16 | 2 | SGD 0.9 1.4e-5, cosine 1.8 | n/a | $\tau$ : $1e^{-4}$ , $\epsilon$ : 0.015 |
| MobileNet-v2 | GraNet | 128 | 180 | 5 | 8 | 1 | SGD 0.9 1e-4 (0.0 for BN), cosine 0.1 | n/a | init density: 0.5 |
|  | STR | 256 | 100 | 5 | 16 | 2 | SGD 0.875 3.751757813e-5, cosine 0.256 | same as the main | init value: -12800 |
|  | OptG | 256 | 180 | 5 | 16 | 2 | SGD 0.9 4-e4, cosine 0.05 | SGD 0.9 0, cosine 0.05 | $\beta$ : 1.0 |
|  | ACDC | 256 | 180 | 5 | 8 | 1 | SGD 0.875 3.05175781e-4, cosine 0.256 | n/a | 8 alterations |
|  | IDP-base/str/optg | 1024 | 180 | 5 | 16 | 2 | SGD 0.9 8e-6, cosine 0.8 | n/a | $\tau$ : $1e^{-4}$ , $\epsilon$ : 0.015 |
| Bert | GMP |  |  |  |  |  | Refer to Section 6 and Table 3 in (Sanh et al., 2020) |  |  |
|  | MOV |  |  |  |  |  | Refer to Section 6 and Table 3 in (Sanh et al., 2020) |  |  |
|  | POFA |  |  |  |  |  | Refer to Section E and Tables 5 and 6 in (Zafrir et al., 2021) |  |  |
|  | STR (85.9%) | 32 | 140 | 5 | 16 | 2 | AdamW 1e-8, multiplicative 1e-4 0.95 | SGD 0.875 0.06, cosine 1e-4 | init value: -90 |
|  | STR (91.7%) | 32 | 140 | 5 | 16 | 2 | AdamW 1e-8, multiplicative 1e-4 0.95 | SGD 0.875 0.065, cosine 1e-4 | init value: -110 |
|  | OptG | 32 | 140 | 5 | 16 | 2 | AdamW 1e-8, cosine 1e-4 | SGD 0.9 0, cosine 1e-4 | $\beta$ : 1.0 |
|  | IDP | 32 | 140 | 5 | 16 | 2 | AdamW 1e-8, cosine 1e-4 | n/a | $\tau$ : $1e^{-4}$ , $\epsilon$ : 0.015 |

**Table 4:** The hyper-parameters in Section 4.

SGD: momentum, weight decay, cosine: learning_rate, AdamW: epsilon, multiplicative: learning_rate, gamma, #wu: the number of warm-up epochs.

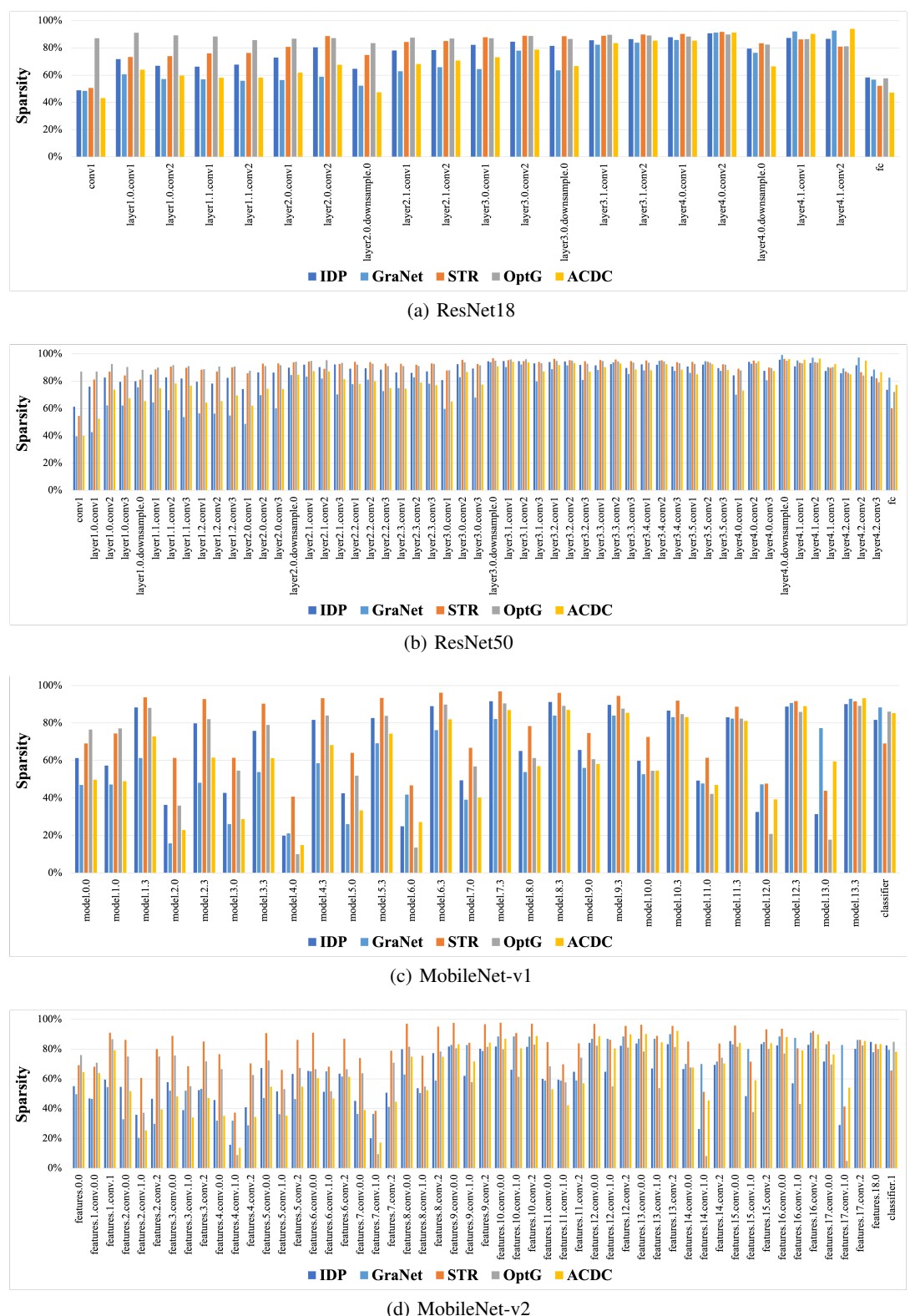

(a) ResNet18

(b) ResNet50

(c) MobileNet-v1

(d) MobileNet-v2

**Figure 5:** Layer-wise sparsity allocation from the experiments in Table 4.

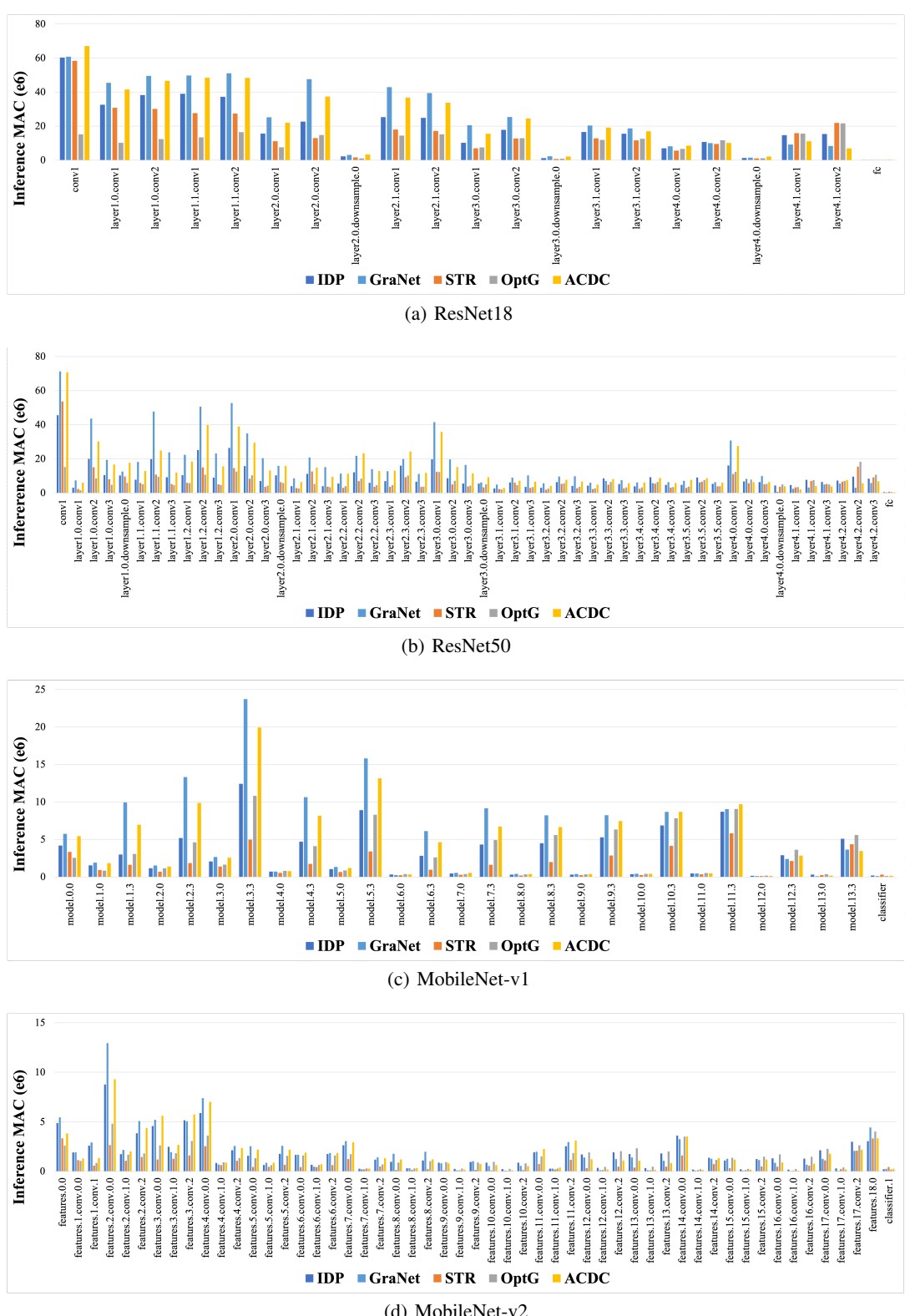

**Figure 6:** Layer-wise Inference MAC distribution from the experiments in Table 4.

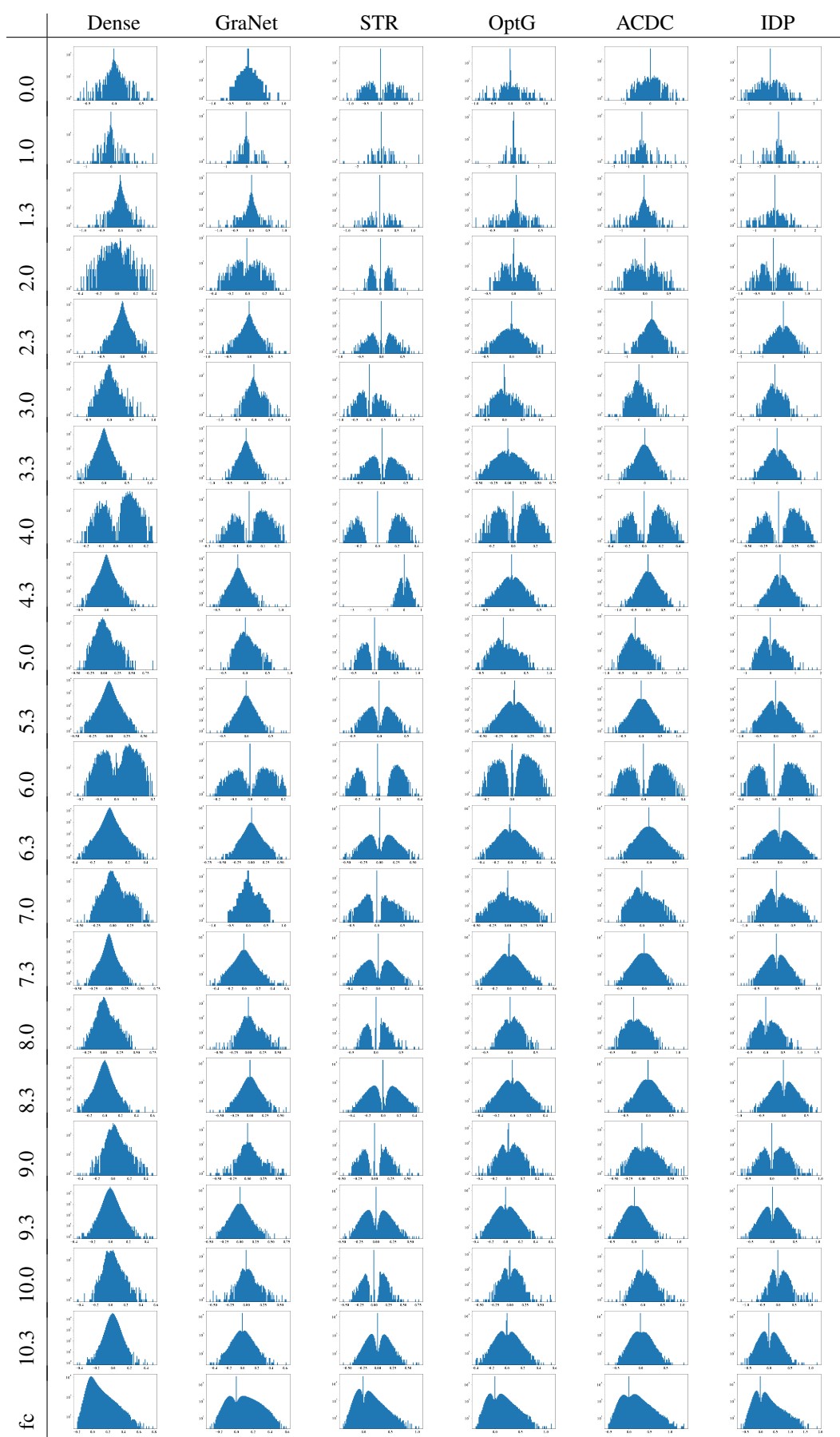

**Table 5:** The weight histograms in log scale for MobileNet-v1 in Table 4.

# A    TRAINING CONFIGURATIONS AND HYPER-PARAMETERS

Since some techniques in Section 4 require extra training parameters and pruning scheduling as shown in Table 1, we disclose the training configurations and hyper-parameters we found the best in Table 4.

# B    TRADE-OFF IN PRUNING.

Pruning for DNN requires exploring a good trade-off between model accuracy and inference latency under a given pruning target. Such a challenge can be elaborated with the MobiletNet-v1 Dense case in Fig. 7 where the following observations can be made:

- The earlier layers have significantly fewer parameters than the later layers while still having comparable inference MACs as shown in (a). For example, the final classifier, which is a linear layer, has the lowest inference MAC but the 2nd largest parameters.

- When per-parameter inference MAC is computed as in (b) (which is in log-scale), we can easily see that the parameters in the earlier layers get much more involved in the inference than those in the later layers. For example, the MAC-per-parameter for the last classifier is only 1.

Then, with a given pruning target, one pruning scheme can favor heavily pruning the classifier, as it is easier to hit the target without affecting model accuracy much (i.e., each parameter shows up only once in the forward pass), but this would fail to reduce the inference MAC enough. Then, the other scheme may favor aggressively pruning the earlier layers to significantly minimize the inference latency at a much greater risk of degrading the model accuracy. Therefore, it is critical to find a good balance between accuracy and inference speed. According to our experimental results, IDP can accomplish such a balance using differential pruning w.r.t. the task loss.

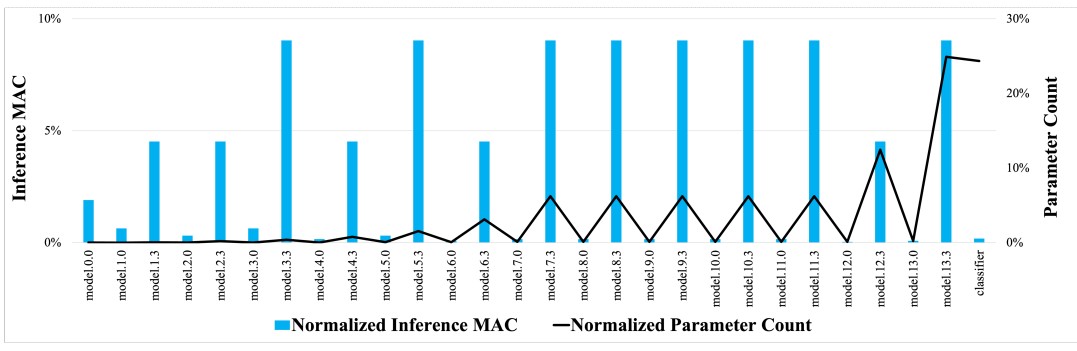

(a) Normalized inference MAC and parameter count for each layer.

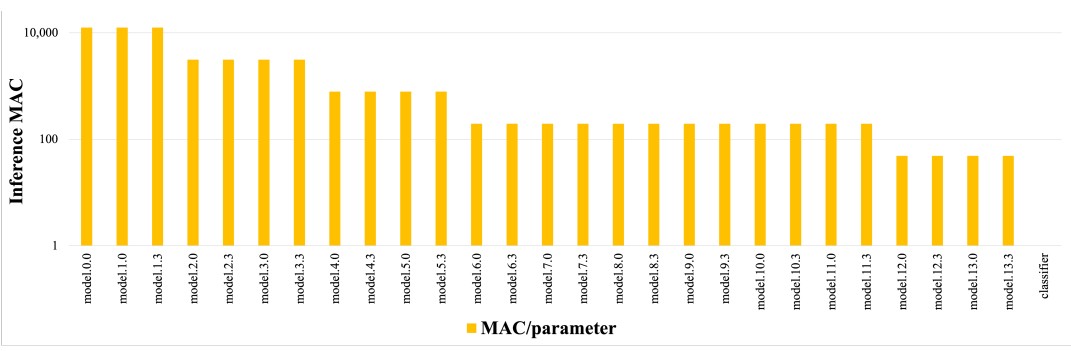

(b) The inference MAC per parameter for each layer.

**Figure 7:** Layer-wise Inference MAC and Parameters from the MobileNet-v1 Dense case in Table 4.

## C   ABLATION STUDY: HYPER-PARAMETER $\tau$ SEARCH

In the current IDP implementation, we use a global $\tau$ to control the level of softness in the pruning mask. Therefore, the selection of $\tau$ affects the model predictive power and should be carefully tuned. In order to explore the methodology for the $\tau$ search, we tried various values for MobileNet-v2 training, and the results are plotted in Fig. 8. The selection of $\tau$ affects the model predictive power as shown in Fig. 8 where there appears to be an optimal $\tau$. For examples of MobileNet-v2, $\tau = 1e - 4$ is the best value and is used for all the experiments in Section 4.

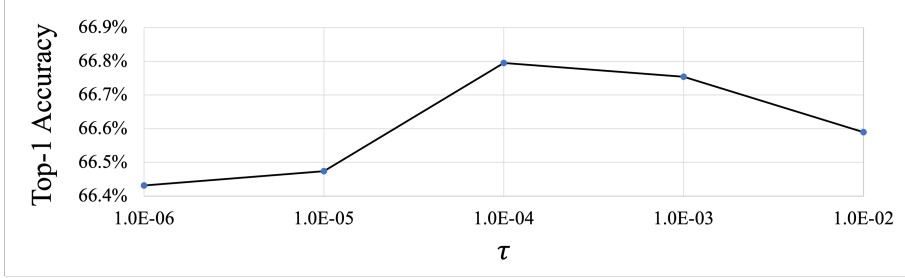

**Figure 8:** MobileNet-v2 with varying $\tau$ values.

Since, Fig. 8 shows a concave curve, one could use a binary search to find the best $\tau$ values w.r.t. the top-1 accuracy. Also, it could be possible to cast $\tau$ as a learnable parameter for each layer or apply some scheduling to improve the model accuracy further (as future work), but still both approaches need an excellent initial point which can be found using a binary search technique.

## D   ADDITIONAL RESULT FOR SECTION 4.

Different approaches made different sparsity allocations per the characteristics of the algorithm for a given pruning target, which results in complex trade-offs between model accuracy and inference speed. We report the detailed sparsity and inference MAC break-down for each layer in Fig. 5 and Fig. 6 on ImageNet1k and summarize our observations as follows:

- **OptG** prunes the early convolution layers quite aggressively in ResNet18 and ResNet50, which leads to very low inference MACs as shown in Fig. 4 (a) and (b), yet at the cost of the worse Top-1 accuracy. For example, the inference MAC of ResNet18 from **OptG** is more than 2x less than that from **ACDC**,

- Interestingly, **STR** becomes aggressive in pruning the early convolution layers in MobileNet-v1/v2, while **OptG** does not expose such behavior to MobileNet-v1/v2 (unlike it did for ResNet18/50). Such characteristics also favor the low inference latency over the model accuracy. Also, **STR** tends not to prune the last linear layer much as discussed in (Kusupati et al., 2020).

- Unlike **OptG** and **STR**, **ACDC** does not prune the early convolution layers much for the tested networks, but prunes somewhat actively for the late convolution and linear layers, which leads to high model accuracies at the cost of higher inference latencies.

- **IDP** is somewhat between **STR** and **ACDC** and modest across all layers in pruning allocation for all the networks, leading to superior accuracy and inference trade-offs. For example, the layers model.13.3 of MobileNet-v1 and features.17.conv.1.0 of MobileNet-v2 have the most difference among algorithms, and **IDP** is modest in pruning these two layers.

- **OptG** has very low inference MACs on the earlier layers of ResNet18 and ResNet50 due to its aggressive pruning on these as seen in Fig. 5 (a) and (b), which leads to the extremely low inference latencies as shown in Fig. 4 (a) and (b).

- **GraNet** tends to prune the earlier layers less but the later layers more in general which explains why **GraNet** shows the highest inference MACs in Fig. 4.

Table. 5 shows the pruned weight histograms of MobileNet-v1 from Table 4. We can observe that each algorithm affects the weight distribution in a slightly different way.

- **STR** prefers to split the distribution more widely than others. For the example of the layer 5.0, **STR** clearly separated the positive and negative weights with a wide gap centered at the zero, while others sis not, except **IDP** created a slight dip around the zero to create mild separation.

- **IDP** tends to spread out the sparsified weight distributions more than others. For the example of the fc layer, the weights from **IDP** range from -0.5 to 2.0, while those from others are from -0.5 to 1.5. On the other hand, **GraNet** tends to keep the weight distributions tight.

We also experimented with varying pruning rates for **IDP, OptG** and **ACDC** for MobiletNet-v1 and ResNet-18 with ImageNet1k, and Bert with MNLI benchmark under the same configurations as in Section 4. Overall, all tested schemes delivered higher accuracy with lower pruning rate, yet we can observe that **IDP** keeps its superiority to other schemes over all the tested pruning rates.

| Network | Method | Sparsity (%) | | | |
|---|---|---|---|---|---|
| | | 80 | 70 | 60 | 50 |
| MobileNet-v1 | IDP | 69.5 | 71.0 | 71.6 | 71.9 |
| | OptG | 68.1 | 69.1 | 69.6 | 69.7 |
| | ACDC | 68.5 | 69.9 | 70.9 | 71.4 |
| ResNet-18 | IDP | 69.8 | 70.8 | 71.0 | 71.3 |
| | ACDC | 69.4 | 70.3 | 70.6 | 70.8 |

**Table 6:** Top-1 accuracy with ImageNet1k: **IDP** outperforms other schemes with various pruning rates.

| Network | Method | Validation dataset | Sparsity (%) | | | |
|---|---|---|---|---|---|---|
| | | | 80 | 70 | 60 | 50 |
| Bert | IDP | matched | 83.7 | 84.0 | 84.3 | 84.7 |
| | | mismatched | 83.4 | 83.8 | 84.4 | 84.5 |
| | OptG | matched | 80.3 | 80.7 | 81.3 | 81.2 |
| | | mismatched | 80.1 | 80.7 | 80.5 | 81.0 |

**Table 7:** Accuracies with MNLI benchmark: **IDP** maintains the similar accuracy lead over other schemes.

## E  PARAMETER-FREE ATTENTION

In this section, we discuss some insight behind our attention and illustrate how to compute the soft-pruned weight matrix following Eq. (4) and Algorithm 3.2.

Our attention is parameter-free on purpose. In pruning, having learnable masks is quite expensive in a way that not only doubles the number of learnable parameters but also may require some particular learning schedulers and tuning, making the overall training flow cumbersome and complicated. Instead, we let weights learn and improve their distribution (weight update changes the distribution naturally) to generate better attention scores for the subsequent iterations in the presence of the current soft mask. Essentially, $t$ captures the current weight distribution, and what would be learned by extra learnable parameters is fused into the weights and their distribution. This way, we can understand where this weight is in terms of overall weight distribution and let backward-pass optimize the weight distribution for better pruning. This is also a key difference from differentiable NAS including DARTS (Liu et al., 2019) where each sub-network comes with a learnable parameter $\alpha$ and the value distribution of the $alpha$ vector drives the search process.

This is feasible due to the structure of the pruning problem. Pruning is optimizing weight distribution for a task loss under the pruning constraints. Updating parameters for task-loss also affects the pruning quality at the same time (i.e., trade-off). Therefore, we need to find a distribution that is best for task loss and also pruning friendly. Hence, instead of having extra parameters, we can directly optimize the weight distribution for high quality pruning. For example, if we surely need $w$ to be

pruned for some reason, instead of having a new parameter to denote 'to-prune', we can actually make $w$ itself smaller relatively against other parameters in the same layer (this is why $t$ is needed) via gradient and weight update. And, IDP is shown to be effective in achieving that with a differentiable soft-mask based on parameter-free attention. Our results confirm that our parameter-free attention is effective by showing superior results to other state-of-the-art algorithms that have explicit mask parameters.

For example, consider a 50% pruning of a weight matrix with $\tau = 0.1$.

$$W = \begin{bmatrix} 0.1 \\ 0.2 \\ 0.4 \\ 0.6 \end{bmatrix}$$

Then, we can compute $t$ to represent the current weight distribution from the pruning perspective: $t = 0.5(0.2 + 0.4) = 0.3$. With $t = 0.3$, we can compute the attention scores to capture the attention to **to-prune** and **not-to-prune** for each weight, which is further dot-producted with $W$ to get the soft-masked weight $\hat{W}$ as follows.

$$\hat{W} = softmax(\frac{concat(0.09 \begin{bmatrix} 1 \\ 1 \\ 1 \\ 1 \end{bmatrix}, \begin{bmatrix} 0.01 \\ 0.04 \\ 0.16 \\ 0.36 \end{bmatrix})}{\tau}) \cdot [0, W] \tag{5}$$

$$= softmax(\frac{\begin{bmatrix} 0.09, 0.01 \\ 0.09, 0.04 \\ 0.09, 0.16 \\ 0.09, 0.36 \end{bmatrix}}{0.1}) \cdot \begin{bmatrix} 0, 0.1 \\ 0, 0.2 \\ 0, 0.4 \\ 0, 0.6 \end{bmatrix} \tag{6}$$

$$= \begin{bmatrix} 0.69, 0.31 \\ 0.62, 0.38 \\ 0.33, 0.67 \\ 0.06, 0.94 \end{bmatrix} \cdot \begin{bmatrix} 0, 0.1 \\ 0, 0.2 \\ 0, 0.4 \\ 0, 0.6 \end{bmatrix} \tag{7}$$

$$= \begin{bmatrix} 0 + 0.031 \\ 0 + 0.076 \\ 0 + 0.268 \\ 0 + 0.564 \end{bmatrix} \tag{8}$$

$$= \begin{bmatrix} 0.031 \\ 0.076 \\ 0.268 \\ 0.564 \end{bmatrix} \tag{9}$$

## F CODE REFERENCES

- **Dense** https://pytorch.org/vision/stable/index.html
- **GradNet** https://github.com/VITA-Group/GraNet
- **OptG** https://github.com/zyxxmu/OptG
- **ACDC** https://github.com/IST-DASLab/ACDC
- **STR** https://github.com/RAIVNLab/STR
- **GMP** https://github.com/RAIVNLab/STR
- **DNW** https://github.com/RAIVNLab/STR

