# OpenReview forum: "IDP: Iterative Differentiable Pruning based on  Attention for Deep Neural Networks"
_ICLR.cc/2023/Conference — Submitted to ICLR 2023_

### Official Review · Reviewer_Dmgj · 2022-10-24

**Confidence:** 4
**Correctness:** 3
**Technical Novelty And Significance:** 3
**Empirical Novelty And Significance:** 3
**Recommendation:** 8

**Clarity, Quality, Novelty And Reproducibility:**

- Equation 4 is a little confusing to me, I would imagine you can swap the position between w and t, then you might also get rid of the negative sign? Is there a specific reason why we cannot do this and have to write this equation in this particular manner?
- You might want to label x and y-xis in your Figure 3b.
- Algorithm 1, line 5, your W is a matrix, are you applying t element-wise? Or is it to each column of W? I would assume this is element wise, then this notation does not look correct.
- I do not understand what is happening to your caption system, or maybe you just missed a caption for Table 1? The organization on Page 8 is simply not visually pleasing.

The rest of the paper is generally clear to me. As I mentioned in weaknesses, soft-mask for pruning is an existing topic, but I do think this paper has provided a new angle to this problem.

**Strength And Weaknesses:**

Strengths

- This paper is well-written with many illustrative plots.
- The design choices are well justified, and the authors have made a detailed comparison to various state-of-the-art fine-grained pruning algorithms.

Weaknesses

- The reported MAC savings are very theoretical, with this pruning granularity, I do not think they transfer directly to run-time performance gains.
- Using a soft-mask for pruning is an existing idea.  However, this paper did perform a nice  reasoning for the usage of a softmax function on these two symbolic states , the proposed method is still an interesting one.

**Summary Of The Paper:**

This paper suggested two major drawbacks on pruning algorithms:

- Difficulties in controlling the sparsity level
- Weights that are pruned away at an early stage do not have a chance to recover

The proposed scheme defines two symbolic states ‘to-prune’ and ‘not-to-prune’, then they generate a soft-mask for values $w$ that sits at the pruning boundaries.

**Summary Of The Review:**

Although I have mentioned about how the evaluation using MACs is not ideal and soft-mask bsed pruning is an existing idea, I do think the proposed symbolic states and its corresponding softmax formulation presents an interesting angle to this pruning problem. So I vote for an accept for this paper.

---

> ### Author Response · Authors · 2022-11-10
> **Response**
>
> We thank you for the reviews and insightful advices to improve our paper.
>
> **Q0: Equation 4 is a little confusing to me, I would imagine you can swap the position between w and t, then you might also get rid of the negative sign? Is there a specific reason why we cannot do this and have to write this equation in this particular manner?**
>
> **A0:** Thank you for the chance to clarify Eq (4). Removing the negative sign and swapping $w$ and $t$ will lead to a form equivalent to Eq (4). Ignoring $\tau$, we can have the following expressions, one from Eq. (4) and the other from your question.
>
> $softmax(-[w^2, t^2])  = [\frac{e^{-w^2}}{e^{-w^2}+e^{-t^2}}, \frac{e^{-t^2}}{e^{-w^2}+e^{-t^2}}]$
>
> $softmax([t^2, w^2])  = [\frac{e^{t^2}}{e^{w^2}+e^{t^2}}, \frac{e^{w^2}}{e^{w^2}+e^{t^2}}]$
>
> If we compare the first elements in the output vectors, we can see that they are essentially identical.
>
> $\frac{e^{-w^2}}{e^{-w^2}+e^{-t^2}} -  \frac{e^{t^2}}{e^{w^2}+e^{t^2}} = \frac{1+e^{-w^2 + t^2} -(e^{-w^2 + t^2}+1 )    }{2+...} = 0$
>
> The second element can be shown to be identical in the same way. Since your suggestion is more concise, we updated Eq (4) and Algorithm 1 with yours. Thank you for the great advice.
>
>
> **Q1: You might want to label x and y-xis in your Figure 3b.**
>
> **A1:** Thanks for raising the missing labels. We have added them to the revised draft.
>
>
> **Q2: Algorithm 1, line 5, your W is a matrix, are you applying t element-wise? Or is it to each column of W? I would assume this is element wise, then this notation does not look correct.**
>
> **A2:** Thanks for the comment. Yes, we are using element-wise. We have improved the notations as follows:
> - We re-wrote $W^2$ into $W \circ W$ to clearly denote element-wise/Hadamard product.
> - We updated the t-side matrix into $t^2 \mathbb{J}$ to improve clarity where $J$ is a matrix of ones.
> - We commented on Algorithms 1 and 2 to clearly highlight the element-wise operations.
>
> **Q3: I do not understand what is happening to your caption system, or maybe you just missed a caption for Table 1?**
>
> **A3:** On our end, the pdf file looks correct and we can see the caption for Table 1. In any case, we will double check the revised pdf file on different computers and softwares.
>
> **Q4: The organization on Page 8 is simply not visually pleasing.**
>
> **A4:** We appreciate your comment. We also found that splitting page 8 into two parts is more readable. We have updated the revised version accordingly.
>
> **Q5: The reported MAC savings are very theoretical, with this pruning granularity, I do not think they transfer directly to run-time performance gains.**
>
> **A5:** We fully agree with the reviewer that the report MAC does not account for the run-time speed up, as MAC only counts the compute parts while the run-time depends on many other complex factors, including memory access, scheduling, and so on. The purpose of repairing MAC savings is to accurately capture the trade-off among accuracy, size, and compute across various algorithms. We have captured your point in Section 4 to improve the clarity.

---

> ### Author Response · Authors · 2022-11-16
> **Follow-up**
>
> Dear Reviewer Dmgj
>
> We like to reiterate our gratitude for your comments and insights. We hope all your comments have been well addressed in the updated draft. If there is additional feedback you may for us, we like to work on that and address it before the rebuttal window closes.
>
> Bests,
> Authors.

---

### Official Review · Reviewer_qrMF · 2022-10-25

**Confidence:** 2
**Clarity, Quality, Novelty And Reproducibility:** 1. **Clarity**
**Correctness:** 3
**Technical Novelty And Significance:** 2
**Empirical Novelty And Significance:** 1
**Recommendation:** 5

**Strength And Weaknesses:**

## Strengths
1. The article is easy to understand.
2. The method is simple and effective and has achieved good performance.

## Weaknesses
### Content
1. In essence, the core contribution of this article is how to choose a threshold t. After the threshold t is obtained, the rest is to use t to generate a mask and then apply the mask to the weight for e2e training. This is actually a very common idea. For example, in the field of NAS, DARTS uses a similar idea. As for gradually adjusting the pruning ratio r, in the field of Detection, it is actually warm up. So these two points are technically feasible, but they lack innovation.
2. I think an experimental result with a low pruning rate r needs to be provided. In this way, the experimental results are complete, proving that this method can work (or at least not deteriorate) under various pruning rates r.

### Writing
1. z(w) is missing a label.
2. Symbol |.| has a different meaning in z(w) and equation 1.

**Summary Of The Paper:**

This post introduces a differentiable pruning algorithm.

The major contributions include:
+ A differentiable and parameter-free pruning algorithm based on attention.
+ Efficiently pruning to offer a high-quality model for a given pruning target.
+ The state-of-the-art results on both computer vision and natural language tasks.

**Summary Of The Review:**

Overall, this article introduces a pruning method that can achieve better performance. But I think this method lacks a certain innovation (refer to the Weakness section). There is still a certain distance from a paper that can be received.

---

> ### Author Response · Authors · 2022-11-10
> **Response 1/2**
>
> Thank you for the feedbacks and the opportunity to revisit our novelty and results. We hope you find our replay and discussion to be satisfactory.
>
> **Q0: In essence, the core contribution of this article is how to choose a threshold t. After the threshold t is obtained, the rest is to use t to generate a mask and then apply the mask to the weight for e2e training. This is actually a very common idea. For example, in the field of NAS, DARTS uses a similar idea. As for gradually adjusting the pruning ratio r, in the field of Detection, it is actually warm up. So these two points are technically feasible, but they lack innovation.**
>
> **A0:** Thank you for the opportunity to highlight the main contribution and novelty of our work. As mentioned by a reviewer, a differentiable NAS (network architecture search) including DARTS relies on a similar technique (i.e., taking softmax) to pick the best sub-network out of many candidates (i.e., a similar decision process), which is briefly mentioned (and DART is also referenced) in Differentiable Pruning of Section 2. We will use mainly DARTS for the explanation purpose in the following discussions.
>
> The main contribution of our work is providing a fresh perspective on how to apply such a decision process WITHOUT involving extra learnable parameters to the pruning problem (please see the last row of Table 1). In the case of DARTS, each sub-network comes with a learnable parameter $\alpha$ as in Eq(2) of the DARTS paper, and the value distribution of the $\alpha$ vector drives the search process. Although having extra learnable parameters is feasible for pruning (for example, OptG in our reference), it is not desirable in practice due to the following reasons:
> - While DARTS may need only a few thousand extra parameters, having extra parameters for pruning is quite expensive. For example, ResNet50 already has 26M parameters, and having a learnable decision parameter for each decision like DARTS means another set of 26M parameters.
> - These extra parameters sometimes require additional optimizers, schedulers, and hyper-parameters, making the training flow complicated (for example, OptG, ACDC, MVP in Table 1).
>
> The novelty of IDP comes from leveraging the structure of the pruning problem. In detail, we let weights learn and improve their distribution (weight update changes the distribution naturally) to generate better attention scores for the successive iterations in the presence of the current soft mask. Essentially, what would be learned by extra learnable parameters is fused into the weights and their distribution. This is feasible, because pruning is essentially optimizing weight distribution with minimal impact on the model accuracy. **For example, if  $w$ is supposed to be pruned for some reason,  $w$ will get smaller relatively against other parameters in the same layer via gradient and weight update without requiring extra parameters to mark 'to-prune'**. And, $t$ represents the overall weight distribution that $w$ can refer to when comparing itself against others in the same layer and adjusting itself for better pruning decision, which is the first attempt in the literature.
> Hence, the weight can represent both the coefficient of DNN and its pruning decision at the same time, thus IDP doesn't need extra $\alpha$ parameter.
>
> Our parameter-free approach sounds simple YET our state-of-the-art results show that it is very efficient/effective and highly practical too. The optimized weight distributions of MobileNet-v1 from IDP (and others) are shown in Table 5 and explained in Section D of Appendix.
>
>
> In detail, DARTS and our method both use a softmax to smooth out the decision function, but in a different way: in DARTS case, the softmax is done across competing candidates (i.e., softmax($\alpha_0$, $\alpha_1$,..), but it is against a $t$ (that captures the weight distribution) and a weight itself in IDP (i.e., to figure out where is this weight in terms of overall weight distribution). The whole pruning now piggybacks on how loss affects the weight update in the presence of a soft mask, and lets weights themselves naturally decide their destination in the subsequent forwards.
>
> We appreciate the reviewer's point w.r.t. NAS which helps us highlight our key contributions and novelty. We have captured your point by updating Section 3.2 and adding Section E to Appendix in the revised draft.

---

> > ### Author Response · Authors · 2022-11-10
> > **Response 2/2**
> >
> > **Q1: I think an experimental result with a low pruning rate r needs to be provided. In this way, the experimental results are complete, proving that this method can work (or at least not deteriorate) under various pruning rates r.**
> >
> > **A1:** We appreciate your comment have performed extra experiments on both vision and NLP tasks, and are happy to share the results under various pruning rates from IDP, ACDC, and OptG (STR is excluded due to the lack of sparsity control). All the hyper-parameters and configurations were kept unchanged from Section 4.
> > |MobileNet-v1 | 80%| 70% |60% |50% |
> > | --- | --- | --- | --- | --- |
> > | IDP | 69.5 | 71.0 | 71.6| 71.9|
> > | ACDC | 68.5 | 69.9 | 70.9 | 71.4 |
> > | OptG| 68.1 | 69.1 | 69.6 | 69.7 |
> >
> > |ResNet-18 | 80%| 70% |60% |50% |
> > | --- | --- | --- | --- | --- |
> > | IDP | 69.8 | 70.8 | 71.0| 71.3|
> > | ACDC | 69.4 | 70.3 | 70.6 | 70.8 |
> >
> > |BERT+MNLI | 80%| 70% |60% |50% |
> > | --- | --- | --- | --- | --- |
> > | IDP | 83.7, 83.4 | 84.0, 83.8 | 84.3, 84.4 | 84.7, 84.5 |
> > | OptG| 80.3, 80.1 | 80.7, 80.7 | 81.3, 80.5 | 81.2, 81.0 |
> >
> > *the 1st number for the matched, the 2nd number for the mismatched.
> >
> > All tested schemes delivered higher accuracy with a lower pruning rate, yet IDP still showed better quality than others. Also, we were able to confirm that IDP exposes no deterioration for the lower pruning rates from 50% to 80%.
> > Thanks to the regularization effects from pruning, Top-1 accuracy  sometimes is better than the dense cases [A, B]. We have added the extra results to the Appendix. Thank you again for suggesting overlooked and essential experiments.
> >
> >
> > **Q2: z(w) is missing a label.**
> >
> > **A2:** We believe that the reviewer refers to the missing labels in Fig. 3 (b), and have added them. The definition of z(w) is in the last paragraph of Page 4. **Please let us know with extra details if we misunderstand your comment.**
> >
> > **Q3: Symbol |.| has a different meaning in z(w) and equation 1.**
> >
> > **A3:** Thank you for the comment $|.|$ in $z(w)$ is the absolute function as $w$ is a scalar, but $|.|$ for matrix $W$ in Eq. (1) returns the number of elements by regarding W as a parameter set. We can see this can cause confusion, thus addressed in the updated draft to avoid confusion by substituting $|W|$ with $n(W)$ with a clear explanation.
> >
> > [A] B. Bartoldson et. al., The Generalization-Stability Tradeoff In Neural Network Pruning, NeurIPS 2022
> >
> > [B] T. Jin et. al., Pruning's Effect on Generalization Through the Lens of Training and Regularization, Arrive 2022

---

> ### Author Response · Authors · 2022-11-16
> **Follow-up**
>
> Dear Reviwer qrMF:
>
> We like to reiterate our gratitude for your comments and insights. We hope our responses on DART vs IDP address your comments.
> If there is anything else we can do to improve our paper, please kindly let us know before the rebuttal update closes. We are looking forward hearing from you.
>
> Bests,
> Authors.

---

> > ### Comment · Reviewer_qrMF · 2022-11-18
> > **Response**
> >
> > Thanks for your reply.
> >
> > What I want to express is that the essence of this practice has been reflected in other methods, and I do not feel that this is an additional contribution.
> >
> > In addition, the structure parameter of DARTS is much smaller than the model, not the same size as the model. As for the question about gradually adjusting the pruning ratio in the first question, it has not been answered. In fact, this is just an engineering parameter adjustment skill, isn't it?

---

> > > ### Author Response · Authors · 2022-11-18
> > > **Follow-up response**
> > >
> > > Thanks for the follow-up.
> > >
> > > Apologize for the comment on the pruning ratio ramp-up. As pointed out, it is similar with a warm-up technique, and it is not our main contribution (no specific novelty claim was made for this as seen at the end of Section 1).
> > >
> > > It is true that the structure parameter of DARTS is much smaller than the model, not the same size as the model. What we like to highlight is that having such structure parameters for pruning is too much overhead (i.e., can be as big as the model size like OptG), so our contribution is around a simple yet effective way to compute a high-quality scoring for two symbolic states without the extra learnable parameters using the context of pruning, supported by the state-of-art results on the initial experiments and the additional experiments with both resnet18, mobilenet_v1, and BERT in our reply (https://openreview.net/forum?id=puguRjbs6Rg&noteId=se9PaKnmN-C).
> > >
> > > Once the right scoring is obtained, using softmax is a common and well-practiced methods, and our claim is not on applying softmax but on how to generate the input to the softmax.
> > >
> > > We hope this has answered to your question and clarify our contribution.
> > >
> > > Thank you again
> > >
> > > Authors.

---

### Official Review · Reviewer_XNYa · 2022-10-28

**Confidence:** 3
**Correctness:** 3
**Technical Novelty And Significance:** 3
**Empirical Novelty And Significance:** 3
**Recommendation:** 6

**Clarity, Quality, Novelty And Reproducibility:**

The paper is clearly written. Although differentiable pruning isn't new, but the use of soft-mask makes the differentiable training simpler - that's a huge plus.

**Strength And Weaknesses:**

The differentiable pruning utilizing soft attention masks results into simplicity and efficiency of training. There is no overhead of auxiliary optimization or additional parameters. As opposed to learning a pruning threshold which is difficult for controlling the sparsity, the proposed method IDP provides higher flexibility and recovery from unwanted pruning in early stages.

**Summary Of The Paper:**

The main contribution of the paper is to devise an iterative differentiable and parameter-free pruning algorithm utilizing attention-based soft pruning masks. They show that the iterative learnable pruning results in improved performance on vision and NLP tasks.

**Summary Of The Review:**

This is a well-motivated and well-written paper. It clearly explains the issues with previous methods, and how they addresses some of those.

---

> ### Author Response · Authors · 2022-11-10
> **Response**
>
> Thank you for the generous comments and we appreciate your pointing out the strength of our approach.
>
> We have revised our draft to highlight our contributions better. Also, more experimental results are added to the Appendix to confirm its performance under varying pruning rates. Please let us know if there is anything we can do to improve our write-up.

---

### Official Review · Reviewer_gR7e · 2022-10-30

**Confidence:** 4
**Clarity, Quality, Novelty And Reproducibility:** They have been presented in the last …
**Correctness:** 2
**Technical Novelty And Significance:** 2
**Empirical Novelty And Significance:** 3
**Recommendation:** 5

**Strength And Weaknesses:**

Strengths:
+ This method is technically clear.
+ The performance on standard benchmarks are quite good。

Weaknesses:
- I think the major issue is presentation. As far as I can see, the method is not iterative, not differentiable and not attention-based. I don't understand what the title means. The presentation makes me so confused and doubt whether I truly understand the technical part.
- By iterative, I thought there are some multi-stage residual formulations like [B] but it turns out not. What does 'iterative' mean in this method? According to algorithm.2, I can only see some standard SGD iterations. If so, shall we call every method 'iterative'?
- By differentiable, I though there are some tricks that allow us to differentiate through some non-differentiable operators but it turns out not. Equations 1-3 are used in a non-differentiable manner. Which part is made 'differentiable'?
- By attention, I though there some learnable attention masks but it turns out that the heuristically assigned mask values are named attention. Of course the authors have the freedom to call it attention. But this is the first time I see the term 'attention' used in this way.
- Finally, since the motivation is to allow dead weights to be non-zero again, an important reference [A] that addresses this old problem is missing.

[A] Dynamic network surgery for efficient dnns, NeurIPS 2016
[B] Network sketching: Exploiting binary structure in deep cnns, CVPR 2017

**Summary Of The Paper:**

This manuscript studies the problem of network pruning for efficiency. The central technical point is a heuristic function (equation.1-3) that assigns a weight-wise soft mask (this is called attention) to each weight during training. The mask value is related to the target sparsity. Combined with a tailored training algorithm that gradually increases the sparsity ratio, the method is shown to out-perform recent pruning state-of-the-art methods. Code is not promised or provided.

**Summary Of The Review:**

Clear technical description (although heuristic), good benchmark results but substantial presentation issues (maybe only to me).

---

> ### Author Response · Authors · 2022-11-10
> **Response 1/2**
>
> Thank you for the comments and the opportunity to improve our presentation. We have improved the draft to capture and address your comments in terms of presentation. The algorithm descriptions and equations have been revised and properly elaborated. We have rearranged your comments based on our reply length.
>
> **Q2: By attention, I thought there some learnable attention masks but it turns out that the heuristically assigned mask values are named attention. Of course the authors have the freedom to call it attention. But this is the first time I see the term 'attention' used in this way.**
>
> **A2:**  Thank you for the comment. We fully understand the reviewer's concern about the presentation, and we have toned down the discussion about attention by substituting it with **parameter-free attention** in the sense that we compute attention without learnable attention masks.
>
> Now, please allow us to discuss the attention more in the context of pruning and our novelty.
>
> Although Wikipedia is not the best source for an academic discussion, attention in machine learning is defined as follows: Attention is a technique that is meant to mimic cognitive attention.
> The effect enhances some parts of the input data while diminishing other parts.
>
> In our case, we use the weight distribution itself as input data (since we prune weights) to enhance or diminish the chance of being pruned for each weight, which complies with the above definition. However, this is a classic definition dating back to 1990, and the following  QKV  attention from Transformers is widespread in modern deep learning.
>
> $Z = softmax( QK) \cdot V$
>
> where QKV from the respective linear layers.
>
> If we re-write our attention formulation in Eq(4) without ignoring the zero (to-prune) part, it can be the following:
>
> $[0, \hat{W}] = softmax( \frac{concat(t^2 J, W \circ W)}{\tau} ) \cdot [0, W]$
>
> Then, we can see $[0, W]$ maps to $V$, and $[0, \hat{W}]$ does to $Z$. Once the attention score is computed, using dot-product ($\cdot$) is the same as well. The difference is that QKV attention uses the scaled-dot product to get the attention score, while ours uses **concat** along with $t$. Hence, there is structure-wise similarity but difference too on how to get the attention score, and we obtain the score without learnable mask parameters. We have added an example of our softmax to Section E of Appendix to help readers.
>
>
> As the reviewer pointed out, we don't have learnable mask parameters.  We could have had learnable mask parameters, but IDP doesn't have them by design. In pruning, having learnable masks is quite expensive in a way that not only doubles the number of learnable parameters but also may require some particular learning schedulers and tuning, making the overall training flow cumbersome and complicated. OptG in our reference is one of such.
>
> Instead, we let weights learn and improve their distribution (weight update changes the distribution naturally) to generate better attention scores for the subsequent iterations in the presence of the current soft mask. Essentially, $t$ captures the current weight distribution, and what would be learned by extra learnable parameters is fused into the weights and their distribution. This way, we can understand where this weight is in terms of overall weight distribution and let backward-pass optimize the weight distribution for better pruning.
>
> This is feasible due to the structure of the pruning problem. In detail, pruning is optimizing weight distribution for a task loss under the pruning constraints. Weight serves as a coefficient of a layer, but also its value can represent the relative chance of being pruned against the other weights in the layer (in IDP, we quantify the chance with the help of $t$).
> Updating parameters for task-loss also affects the pruning quality at the same time (i.e., trade-off). Therefore, we need to find a distribution that is best for task loss and also pruning friendly. Hence, instead of having extra parameters, we can directly optimize the weight distribution for high quality pruning.
>
> **For example, if  $w$ needs to be pruned for some reason, instead of having a new parameter to denote 'to-prune', SGD will actually make $w$ itself smaller relatively against other parameters in the same layer via gradient and weight update**. And, IDP is shown to be effective in achieving that with a differentiable soft-mask based on parameter-free attention. Our results confirm that our parameter-free attention is effective by showing superior results to other state-of-the-art algorithms that have explicit mask parameters.
> FYI, the optimized weight distributions of MobileNet-v1 from IDP (and others) are visualized in Table 5 and explained in Section D of Appendix.
>
> We thank the reviewer again for bringing up an important aspect of our work, and we have summarized the above discussion in Section E of Appendix.

---

> > ### Author Response · Authors · 2022-11-10
> > **Response 2/2**
> >
> >
> > **Q0: By iterative, I thought there are some multi-stage residual formulations like [B] but it turns out not. What does 'iterative' mean in this method? According to algorithm.2, I can only see some standard SGD iterations. If so, shall we call every method 'iterative'?**
> >
> > **A0:** Thank you for the comment. The iterative part is that IDP revisits the soft-mask generation every epoch with a different pruning target as in line 10 of Algorithm 2. And, within an epoch, the soft-mask values are iteratively updated to accommodate the updated weight values. The iteration in IDP is aligned with the SGD iteration on purpose, not to add extra loop overheads on the forward/backward pass. The goal of this iterative operation is elaborated in the first paragraph of Section 3.3.
> >
> >
> > **Q1: By differentiable, I thought there are some tricks that allow us to differentiate through some non-differentiable operators but it turns out not. Equations 1-3 are used in a non-differentiable manner. Which part is made 'differentiable'?**
> >
> > **A1:** Thank you for the question and the opportunity to clarify it.
> >
> > - The reviewer is correct that Eq. (1,2,3) are used in a non-differentiable matter, but Eq. (1,2,3) don't need gradient back-propagation as getting $t$ for Eq(4) is the sole purpose. Hence, in our implementation, we excluded  Eq. (1,2,3) from autograd (using torch.no_grad()), making $t$ appear as a scalar value during forward/backward pass. We made this clear in Section 3.2, and added a comment to Algorithm 1 as well.
> >
> > - The part we made differentiable is how to generate the pruning mask which corresponds to Eq(4). The conventional way of generating the pruning mask using a threshold alone is a non-differentiable operation. For example,
> > $m(w) = 1$ if $w \ge t$ and $m(w) = 0$ if $w<t$, which is not differentiable. In our method, we propose to generate the mask itself in a differentiable way using the boundary point $t$ by softening the mask itself with the softmax function as in Eq. (4).
> >
> > By combining the above two, our end-to-end forward/backward passes remain differentiable. Please note that a conceptually similar paper is referred in the Differentiable Pruning in Section 3 [A]. In this work, the non-differentiable mask generation based on the threshold became differentiable by using the first derivative of foothill function. We have emphasized the above discussion in the revised draft to improve clarity. Thank you again for you valuable input.
> >
> > **Q3: Finally, since the motivation is to allow dead weights to be non-zero again, an important reference "Dynamic network surgery for efficient dnns, NeurIPS 2016 " that addresses this old problem is missing.**
> >
> > **A3:** Thank you for pointing out a relevant paper. We have added the paper to the discussion in Section 3: The mentioned paper proposes iterative pruning and splicing steps where the splicing will reverse the pruning decision based on the weight importance. The weight importance is determined by the magnitude of the weight and two user-provided thresholds as in Eq (3) of the paper. A similar yet more advanced approach is ACDC referred in our submission where the reverse pruning decision is made by IHT (iterative hard thresholding) algorithm.
> >
> > **Q4: Code is not promised or provided.**
> >
> > **A4:** Thank you for bringing up an important topic. At the time of submission, it was unclear to us when the code can be open-sourced. Now, we have a tentative schedule to release IDP to the public in late June 2023 with detailed documents. Hope this can be a good contribution to the community.
> >
> > - [A] Ramchalam Kinattinkara Ramakrishnan, Eyyüb Sari, and Vahid Partovi Nia. Differentiable mask for pruning convolutional and recurrent networks. In International Conference on Computer and Robot Vision 2020

---

> ### Author Response · Authors · 2022-11-16
> **Follow-up**
>
> Dear Reviwer gR7e:
>
> We like to express our appreciation for your comments and feedbacks again. We wonder if you can further provide feedbacks on our responses to your original reviews. Please kindly let us know if you have any further questions on your work. We are more than happy to work with you to address your comments before the rebuttal update closes.
>
> Bests,
> Authors.

---

### Author Response · Authors · 2022-11-10
**Response**

We thank all the reviewers for their thorough comments and their many suggestions for improving the exposition of the paper. We have updated the paper with the following main changes:

1. We ran additional experiments with various pruning rates to demonstrate the robustness of our method. The results are added to Appendix.
2. We added more discussion on the benefits of our approach in Section 3.2  along with the newly added Section E in Appendix, focusing on its similarity and difference from the existing approaches.
3. We improved the notations in equations and algorithms to improve readability and clarity.
4. We broke page 8 into one set of graphs and one table, and placed them on different pages to improve readability.
5. We added the missing dense baseline to Table 2 for the NLP experiments.
6. We added the missing labels for Fig. 3 (b).
7. We fixed typos and cleaned up the reference format.
8. We slightly modified the title to highlight the novelty and contributions.

---

### Decision · Program_Chairs · 2023-01-20

**Decision:**

Reject

**Justification For Why Not Higher Score:**

After substantial discussion, two reviewers still lean reject; the AC concurs that there are major issues with presentation and clarity that need to be resolved before this paper is fit for publication.  These are severe enough to outweigh some promising results.

**Justification For Why Not Lower Score:**

N/A

**Metareview: Summary, Strengths And Weaknesses:**

This paper develops a new network pruning algorithm in which a soft-mask, which itself is a dynamic function of the weight matrix, acts as the pruning mechanism.  The paper highlights this mechanism as being differentiable and parameter-free (excepting hyperparameters governing the weight to mask function), while experimentally yielding efficient pruned networks across a range of architectures and tasks.

After the author response and discussion, reviewer opinion was split.  The AC prompted reviewers for additional feedback in a private discussion thread and opinion remained split, with reviewers reaffirming their publicly stated arguments both for and against the paper.  The AC has taken a detailed look at the paper, author responses, and discussion, and concurs with reviewers on the following points:

(1) As argued by Reviewer gR7e, it does not appear appropriate to frame the method using the terminology of "attention".  A choice between two states (prune or do not prune) does not match the context in which "attention" is used by the community: that of dynamic selection, transformation, and remixing of information over a large set of inputs.  The fact that an equation contains a dot product and a softmax does not suffice; classifiers existed prior to the introduction of attention-based neural architectures.  Latching onto and misusing a popular term is not in keeping with the standards of presentation clarity expected of an ICLR paper.  In further discussion with the AC, Reviewer gR7e characterizes the presentation as "potentially misleading usage of technical terms"; the AC agrees.

(2) The proposed technique is not novel in terms of being the first end-to-end differential approach to pruning.  Reviewer qrMF points to DARTS as being a similar idea in the context of NAS.  The authors themselves mention the "Differential Mask for Pruning..." paper of [Ramchalam et al., 2020].  Note that another example of possibly relevant prior work in this regime (not mentioned in the discussion) is the "Continuous Sparsification" approach of [Savarese et al., 2020].  After discussion, Reviewer qrMF remains concerned about the overall novelty of contributions.

(3) There may be novelty in the parameter-free aspect of the approach (not relying on auxiliary mask variables).  Though, there are also concerns raised by Reviewer gR7e about the heuristic nature of the technique.

(4) Experimental results demonstrate consistent, though perhaps incremental in magnitude, improvements due to the proposed method over an array of prior work on pruning.  The AC agrees with Reviewer Dmgj and others that this is a strength of the paper.

However, as a whole, the concerns over presentation (1,2), novelty (2), and heuristic aspects of the work (3) are serious enough that the AC cannot recommend acceptance in its current form, despite the promising experimental results (4).  The choice of presenting the approach as "attention" is either entirely misleading or insufficiently explained so as to appear misleading.  In either case, the paper needs a major rework and additional review to address this issue (1), which is the most severe problem.  The paper might also benefit from additional clarification on which aspects are novel with respect to prior approaches, and ablation analysis of how parameter-free masks directly compare to parameterized masks and/or the design space of the heuristic for mapping weights to masks.